# Timeliness and missed opportunities for vaccination among children aged 0 to 23 months in Dschang health district, West region, Cameroon: A cross-sectional survey

**Arsène Gautler Dombou Zeufack**[1], **Diomede Noukeu Njinkui**[1], **Solange Whegang Youdom**[1]*, **Jérôme Ateudjieu**[1,2]

**1** Department of Public health, Faculty of Medicine and Pharmaceutical Sciences, University of Dschang, Dschang, Cameroon, **2** Division of Health Operations Research, Ministry of Public Health, Yaoundé, Cameroon

* swhegang2002@yahoo.fr

**Data Availability Statement:** The main dataset is being explored for secondary analysis. However,

## Abstract

Missed opportunities for vaccination (MOV) reflect quality of immunization service. The objective of this study was to assess vaccination timeliness, prevalence, and characteristics of MOVs among children aged 0–23 months, as well as knowledge, attitude and practice of health workers towards immunization. An exit interview method was used to select caregivers and health personnel. Selection took place in 26 health facilities within 14 health areas in the Dshcang Health district. Data were collected using two face-to-face questionnaires adapted from the World Health Organization (WHO) tools. We conducted an evaluation of all free vaccines in the Expanded Programme on Immunisation (EPI). We studied timeliness, assessed MOV, and knowledge, behaviour and attitude of health workers on immunization. Basic statistical tests were used to study the association between MOV and socio demographic characteristics. A total of 363 children aged 0 to 23 months were surveyed. A total of 88 (91.66%) of health personnel agreed to participate in our study. A total of 298 (82.1%) children had vaccination cards with dates, leading to 18% not completely vaccinated. Vaccination timeliness ranged from 20% to 77%. Overall MOV estimated was 23.83%, range from 0% to 16.4% among all vaccines. Among health workers, 70.45% (62/88) had insufficient knowledge on vaccination, 73.86% assessed the vaccination status of children during any routine visit and 74% ask parents to bring the child's vaccination record to any health facility visit. The study highlighted presence of MOV among children. Strategies for remedying this includes strengthening parents' knowledge, organizing refresher courses for health workers on vaccination, and systematically assessing children's vaccination status.

## Introduction

Immunization is one of the most effective health interventions for protecting children against vaccine-preventable diseases (VPD) [1]. It involves the immunization of individuals against a

we uploaded the minimal dataset necessary to replicate the quantitative findings of this study, as well as the R code used, as Supporting Information files.

**Funding:** The authors received no specific funding for this work.

**Competing interests:** The authors have declared that no competing interests exist.

specific infectious disease through the administration of a vaccine [1]. Indeed, it prevents children from suffering serious physical, mental or neurological disabilities [1]. Vaccination has been shown to control and eliminate life-threatening infectious diseases, and it is estimated that 2–3 million childhood deaths per year are prevented by vaccination, yet 19.7 million children under one year of age do not receive basic vaccines [2]. In Cameroon, vaccination has been offered throughout the country since 1982 through the EPI of the Ministry of Public Health, which targets children aged 0 to 11 months and pregnant women [3].

Recent assessments conducted by the World Health Organization (WHO) on the magnitude of missed opportunities for vaccination (MOV) in Africa indicated that 96% of children attending health facilities for health care respectively came out without receiving the doses of vaccines for which they were eligible [4]. Since 2006, global immunization coverage has remained constant at 80%, meaning that one-fifth of the global birth cohort is not vaccinated with DPT3. MOVs are in most cases due to non-compliance with established policies and procedures in the delivery of immunization services [4]. This highlights the burden of MOVs on immunization programs in some African countries where MOVs have been associated with the failure to fully immunize children eligible for national immunization programs, thus contributing to the partial achievement of national immunization coverage targets [5–7]. Despite multiple evidences of EPI effectiveness, vaccine coverage remains suboptimal, and missed opportunities for vaccination have been repeatedly identified as contributors to poor immunization performance [8–13].

In order to provide information that could contribute to the improvement of child health, access to immunization services and the achievement of optimal immunization coverage, this study aims to study vaccination timeliness and estimate the prevalence of missed vaccination opportunities among children aged 0–23 months in the Dschang Health District. In addition, we aim to assess the knowledge, behaviour/attitude, and practice (KAP) of health workers on immunization.

## Methods

### 1. Study design

We used a descriptive cross-sectional study targeting children aged 0–23 months and health personnel in the Dschang health district. An exit interview methodology was used for data collection. Two structured questionnaires were administered face-to-face to caregivers and health personnel in 26 health facilities in 14 health areas of the district. These questionnaires were adaptations of the World Health Organization (WHO)'s tools for assessing missed opportunities for vaccination [4]. Collected data were subsequently managed using Excel 2016, Epi Info version 7.2 and Stata 16.

### 2. Selection of study participants

All children aged 0–23 months who came health facilities on the day of the interview, without restriction of the reason for the visit and health personnel (in the preventive and curative services), were eligible for this study. A questionnaire assessing knowledge, attitudes and practices of vaccination was administered to 10 health personnel by health facility.

### 3. Sample size estimation

Assuming a prevalence of MOV of 65.7% [9], with a confidence limit of 95% and a precision 5%, led to a sample of 345.3. To minimize the number of non-respondents and assuming 15% of non-respondents, we obtained a final sample of 406 children aged 0 to 23 months. The sample size used for the KAP study, estimated from a quota sampling, was 203 health workers.

## 4. Collection tools and data collection

Questionnaires were designed on the KoBoToolbox platform and administered via smartphones and tablets. Free and informed consent was obtained from all participants prior to beginning of data collection. The questionnaires were administered to caregivers and health personnel using the KoBoToolbox mobile app and were anonymous to maintain confidentiality. These data were collected from caregiver statements and children's immunization records for the exit interviews and from health staff statements for the KAP survey. The dates and sources of immunization were extracted from the immunization records. If the planned number of interviews was not obtained in one morning in the health facility, another with the same characteristics (type, size, and category) was requested to compensate.

## 5. Data management and analysis

We described all surveyed participants (children, caregivers, and health workers). Knowledge, attitude, and practice of health workers towards vaccination, were described using frequencies. For vaccination timeliness, analysis was carried on children with report cards seen, that contain at least one vaccination date. We classified vaccination timeliness as "early," "timely," or "late" based on the child's date of birth, the age and interval for each vaccine dose according to the national vaccination schedule. We computed an overall MOV prevalence based on the same sample, and use participants' characteristics to study the association with MOV occurrence. We also looked at the MOV for each antigen. All vaccines present in the immunization schedule at the time of the evaluation, except for the birth dose of viral hepatitis B (Low numbers of children vaccinated with HepB0 because vaccination is not routine.) vaccine and the second dose of measles vaccine (The number of children who received RR2 in our sample was very small to be representative of the study population.), were included in the MOV analysis. Children were considered to have MOV if they met the following four criteria: i) dates of vaccination were correctly documented, ii) they were not up to date according to the recommended national immunization schedule at the start of the visit, iii) they reported no contraindication to vaccination on the day of assessment, and iv) they left health facility without receiving one/ several doses of vaccine for which they were eligible. Timeliness and MOV analysis were ran under the free statistical package R version 4.0.3.

## 6. Ethical considerations

The study obtained an approval from the National Ethics Committee for Research in the Humanities of Cameroon N˚2021/09/94/CE/CNERSH/SP. In addition to this, the study obtained an authorization from the Head of the Dschang health district, Heads of the health facilities, and free and informed consent of children' caregivers and health personnel.

# Results

At the end of the data collection, 390 caregivers and 96 health workers were surveyed. Among them, 382 caregivers and 88 health personnel agreed to participate in the study, i.e., participation rates of 97.95% (382/390) and 91.66% (88/96) respectively. Data were collected in 26 health facilities belonging to 14 health areas of the district. The interviews took place in 26 (including 2 additional) district health facilities that offer immunization services, including 17 Public and 11 private health facilities (S1 Fig). Flow chart of the surveyed participants and the sample size used in the computation of timeliness and MOV indicators, is presented on Fig 1.

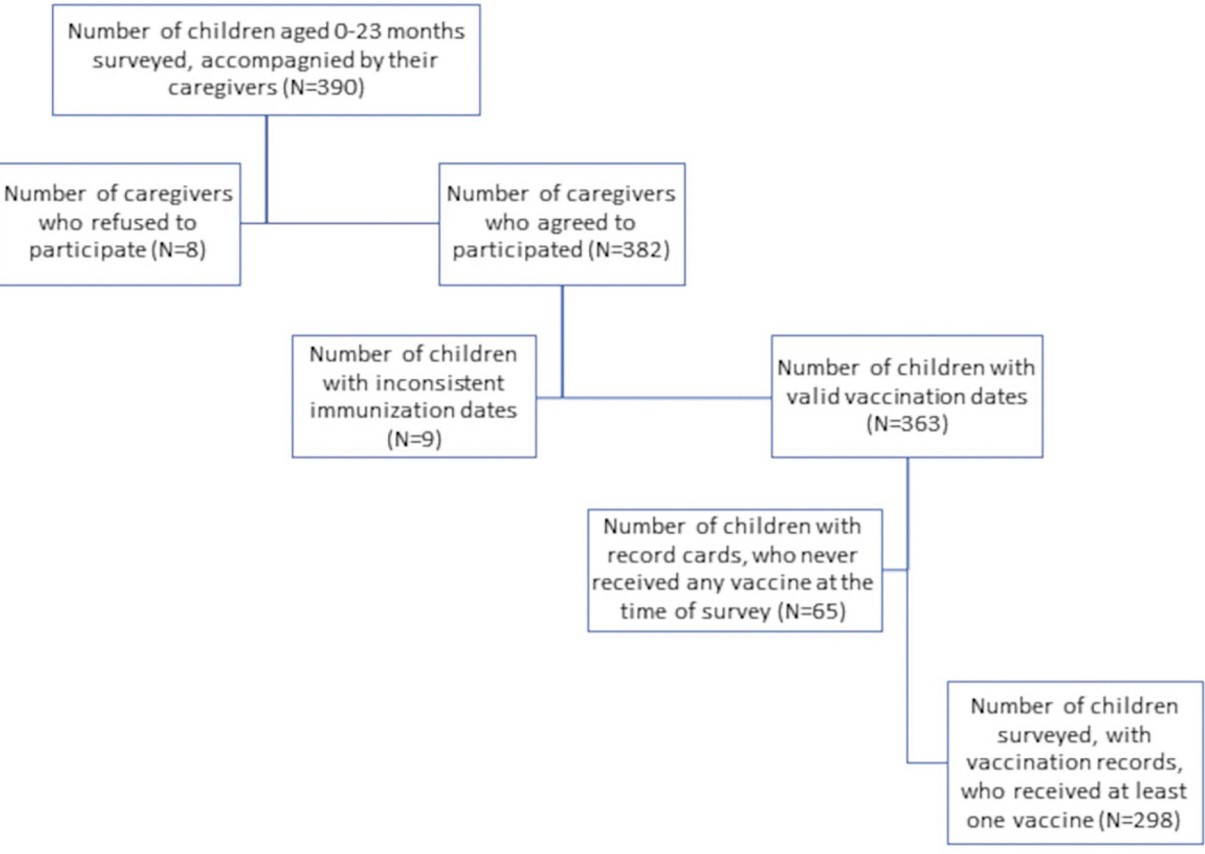

**Fig 1. Flow chart for the identification of MOVs in the Dschang health district in 2021.**

## 1. Socio-demographic characteristics of study participants

Of the 363 children with a record of vaccination dates included in the analyses (Table 1), 50.1% (182/363) were female and their age ranged from 0 to 22 months, i.e., an average of 4.5 ±4 months, and the [0–11] month age group was predominant (92.36%). Of these children, 28.9% (105/363) were first births and 48.21% (175/363) were at least the third birth. In our sample, 1.93% (7/363) of eligible children had not yet been vaccinated. A total of 96.7% (351/363) of the participants were the mothers of these children (Table 1), their age ranged from 14 to 60 years with an average of 28±6 years. Among them, 69.97% (254/363) had a secondary education, 27.82% (101/363) were housewives and 62.3% (226/363) lived in urban areas compared to 37.7% (137/363) in rural areas.

## 2. Card possession and vaccination sources

Of 373 children, 363 had their immunization booklet in the health facility on the day of the interview, i.e., a booklet possession rate of 97.32%. Types of vaccination were recorded as: vaccination records, caregivers recall, tick mark, and not vaccinated child (S1 Table). Among those with cards, 298 (82.1%) had at least one vaccination date for the following vaccines: BCG, OPV0, MCV1, YF, ROTA1, ROTA2, PENTA1, PENTA2, PENTA3, OPV1, OPV2, OPV3, PCV1, PCV2, PCV3, IPV; suggesting 18% not completely vaccinated. Data sample used to compute vaccination timeliness and missed opportunity indicators are found in S1 Data. In addition, a piece of R code used for the computation can be found in S1 File.

**Table 1. Socio-demographic characteristics of surveyed participants (caregivers and children aged 0 to 23 months) in the district in 2021.**

| Variables | Caregivers (n = 363) | | | Children (n = 363) | | |
|---|---|---|---|---|---|---|
| | Modalities | Numbers | Proportion (%) | Modalities | Effectives | Proportion (%) |
| **Âge class** | [15–26] years | 181 | 49.8 (181/363) | [0–11] months | 336 | 92.6 (336/363) |
| | [27–31] years | 97 | 26.7 (97/363) | [12–23] months | 27 | 7.5 (27/363) |
| | [32–63] years | 85 | 23.4 (85/363) | - | - | - |
| **Sex** | Female | 362 | 99.7 (362/363) | Female | 182 | 50.1 (182/363) |
| | Male | 1 | 0.27 (1/363) | Male | 181 | 49.9 (181/363) |
| **Birth rank** | - | - | - | 1rst | 105 | 28.9 (105/363) |
| | - | - | - | 2nd | 83 | 22.9 (83/363) |
| | - | - | - | ≥3rd | 175 | 48.2 (175/363) |
| **Children immunized** | - | - | - | Yes | 356 | 98.1 (356/363) |
| | - | - | - | No | 7 | 1.93 (7/363) |
| **Relationship between caregiver and child** | Mother | 351 | 96.69 (351/363) | - | - | - |
| | Grandmother | 7 | 1.93 (7/363) | - | - | - |
| | Other (father, sister, aunt) | 5 | 1.37 (5/363) | - | - | - |
| **Level of education** | Primary | 38 | 10.5 (38/363) | - | - | - |
| | Secondary | 254 | 70.0 (254/363) | - | - | - |
| | Higher | 71 | 19.6 (71/363) | - | - | - |
| **Profession** | Hairdresser | 50 | 13.8 (50/363) | - | - | - |
| | Seamstress | 43 | 11.8 (43/363) | - | - | - |
| | Schoolgirl (pupil + student) | 65 | 17.91 (65/363) | - | - | - |
| | Housewife | 101 | 27.8 (101/363) | - | - | - |
| | Farmer | 19 | 5.2 (19/363) | - | - | - |
| | Teacher | 41 | 11.3 (41/363) | - | - | - |
| | Other | 44 | 1.1 (44/363) | - | - | - |
| **Area of residence** | Rural | 137 | 37.7 (137/363) | Rurale | 137 | 37.7 (137/363) |
| | Urban | 226 | 62.3 (226/363) | Urbaine | 226 | (226/363) |

## 3. Timeliness of vaccination

Among those with vaccination records and dates, on-time vaccination ranged from 42% (OPV0, n = 278) to 76% (PCV1, n = 196). All multiple doses have vaccination timeliness up to 75% (Fig 2, S2 Table). In addition, there was a non-neglected proportion of early doses. Indeed, 15.82% of ROTA1 vaccines were received early, and 17.77% for OPV1 (Fig 2, S2 Table).

## 4. MOV indicators according to valid vaccination dates

The prevalence of MOV for any vaccine was recorded as 23.83% (71/298), with a 95% confidence interval (CI) of 19.34%-29% (Table 2). This percentage indicated children with vaccination records that contained valid dates, aged 0 to 23 months, who were eligible to one or more antigens at the time of survey but, who finally did not receive the vaccines for which they were age eligible. Raw percentage of MOV (all vaccination sources: cards, recall, and tick mark) was 47.6% (142/298) (data not tabulated).

When analysis was cross-tabulated by each vaccine, results showed prevalence that varied from 0% (opv2-3, pcv2-3, mcv1) to 16.47% (bcg) (Table 2). Indeed, among 249 children who were eligible for bcg, 41 (16.74%) experienced an MOV for bcg. In addition, 5.3% (11/206) experienced MOV for rota1, 6.37% (13/204) for penta1, 9.23% (12/130) for ipv. Overall, except for bcg, MOV for each dose was less than 10% (Table 2).

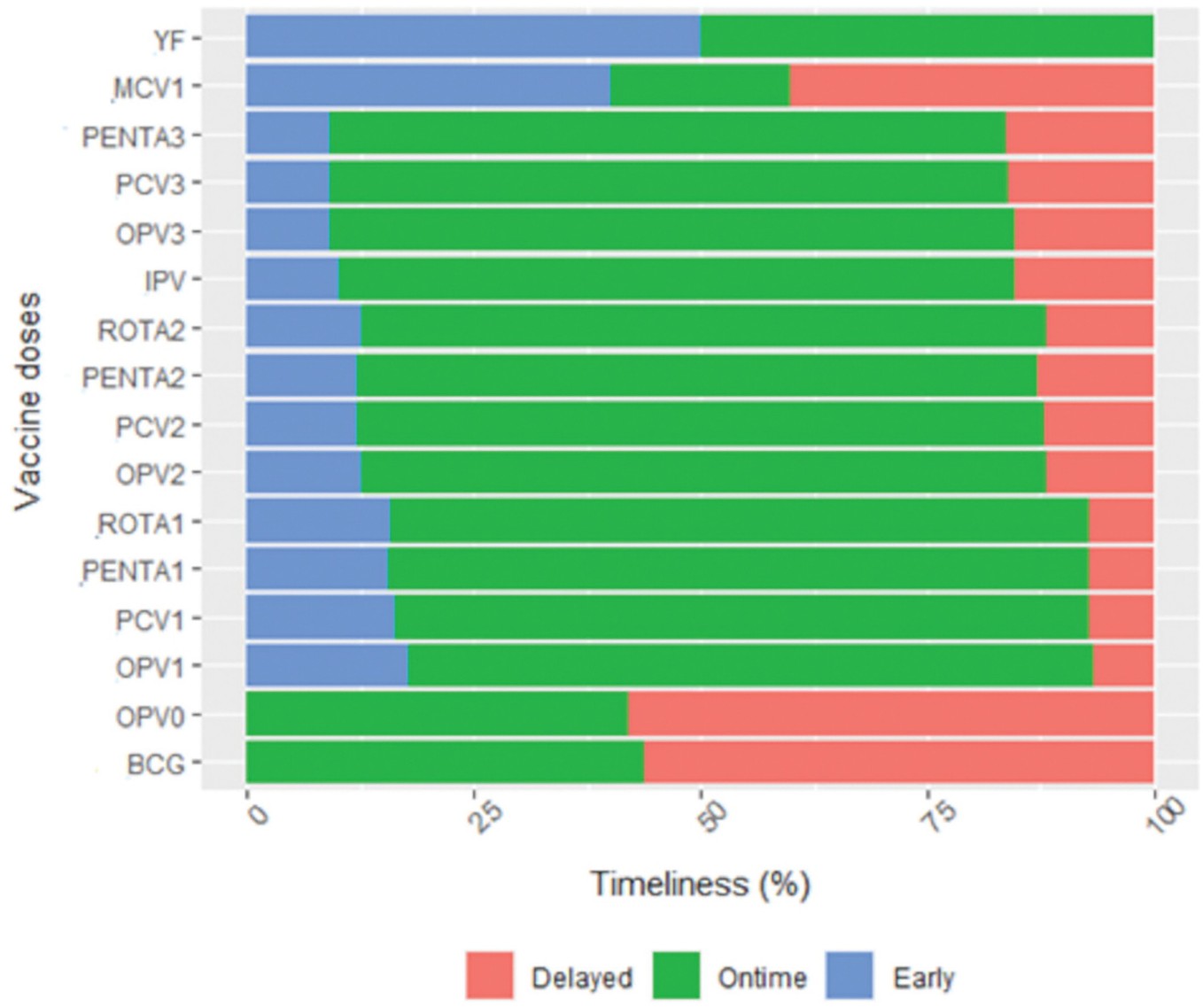

**Fig 2. Vaccination timeliness among children aged 0 to 23 months in the Dschang health district in 2021.**

**Distribution by socio-demographic characteristics of missed opportunities for vaccination in the Dschang health district.** Table 3 presents the distribution of missed immunization opportunities according to the socio-demographic characteristics of the caregivers and the children. Among children surveyed in the health facility, with cards seen and vaccination dates, 25.4% experienced MOV for any vaccines in the health centre, and 49.3% at the district hospital level. There was a positive association between type of health facility and MOV (p-value = 0.009) i.e., MOV was more elevated in the public sector (74.6%) compared to the private sector (25.4). There was a borderline association between caregiver's age and MOV (p-value = 0.06).

## 5. KAP study of health personnel in the Dschang health district

Sociodemographic information of health workerswas presented on the S3 Table. Overall, 69.05% (59/88) were employed in the public sector, 75% (66/88) were women and 56.82% (50/

**Table 2. Prevalence of MOV by vaccine dose in children aged 0–23 months in the Dschang health district, 2021.**

| Antigens | #Doses administered* | Min-Max age at vaccination | Median age at vaccination | Elligible children+ | Prevalence (%) |
|---|---|---|---|---|---|
| BCG | 249 | 0–398 | 11 | 249 | 16.47 |
| OPV0 | 283 | 0–219 | 11 | 117 | 5.98 |
| OPV1 | 216 | 0–415 | 47 | 206 | 5.34 |
| Rota1 | 215 | 6–289 | 47 | 206 | 5.34 |
| PCV1 | 215 | 6–289 | 47 | 206 | 6.31 |
| Penta1 | 213 | 6–289 | 47 | 204 | 6.37 |
| OPV2 | 173 | 43–378 | 78 | 168 | 0 |
| Rota2 | 172 | 43–317 | 78.5 | 168 | 0.6 |
| PCV2 | 172 | 43–317 | 78 | 167 | 0 |
| Penta2 | 172 | 43–443 | 78 | 168 | 0.6 |
| OPV3 | 138 | 74–345 | 110 | 127 | 0 |
| IPV | 138 | 46–345 | 110 | 130 | 9.23 |
| PCV3 | 138 | 74–317 | 110 | 126 | 0 |
| Penta3 | 138 | 74–345 | 110 | 125 | 0.8 |
| MCV1 | 68 | 183–506 | 281.5 | 57 | 0 |
| YF | 64 | 183–343 | 280 | 54 | 1.85 |
| Overall | | 0–22 | 3 | 298 | 23.83 (19.34–29)++ |

Abbreviations: **BCG**: Bacillus Calmette Guerin; **OPV**: oral polio vaccine; **Penta**: Diphtheria-tetanus-pertussis-hepatitis B Haemophilus influenzae type b; **PCV**: pneumococcal conjugated vaccine; **Rota**: rotavirus; **MCV1**: measles-rubella first dose, **YF**: yellow fever.

*Number of children vaccinated at the time of survey including all sources (record card, recall, tick mark)

+number of eligible children i.e. children who reached the recommended age of vaccination, who received the vaccine within the window of eligibility; ++ prevalence of MOV for simultaneous vaccines and its 95% confidence interval.

88) were nurses. The age ranged from 20 to 55 years with an average of 34±9 years and the length of time in the profession ranged from 4 months to 32 years. In our sample, 60.23% (53/88) were employed in preventive services (S3 Table).

**a. Knowledge of health workers on vaccination.** In our sample, 13.64% (12/88) said they had never received basic training on vaccines and control of VPD (Table 3). Of the 86.36% (76/88) who had received training, it had taken place during basic training and 78.41% (69/88) had already received refresher training or attended a seminar on immunization. Vitamin A was identified by 72.62% (61/88) of the health workers as one of the EPI vaccines, while 45.71% (16/88) did not identify any of the vaccines offered.

Regarding the knowledge of contraindications to vaccination, only 40.91% (36/88) chose "pneumonia and other serious diseases" as the correct answer (Table 4). The majority (70.45%) of health workers (62/88) reported insufficient or outdated knowledge about immunization and 28.41% (25/88) reported that filling out immunization documents was a cause of delay in providing routine immunization services on the day of immunization.

**b. Attitudes of health personnel towards vaccination.** Only 73.86% (65/88) of the health workers considered that the child's immunization status should be systematically assessed at each visit to the health facility (S4 Table), while 25% (22/88) of them thought that it should be done by the child's parents and 42.04% (37/88) by the health workers in the consultation and hospitalization departments. Three quarters (76.47%) of the health workers (65/88) identified the negative beliefs of the children's attendants, the long distances between the homes and the health facility (52.94%; n = 45/88) and the attendants' schedules that were incompatible with the vaccination times (51.76%; n = 44/88) as the reasons for the failure of full vaccination (S4 Table).

**Table 3. Distribution of MOVs among children aged 0 to 23 months in the Dschang health district in 2021.**

| Variables | Modalities | Child have MOV | | |
|---|---|---|---|---|
| | | Yes (n = 71) | No (n = 227) | P-value** |
| **Health facilities** | Health Centre | 18 (25.4) | 97 (42.7) | 0.0000 |
| | District Medical Center | 5 (7) | 27 (11.9) | |
| | Integrated Health Center | 13 (18.3) | 70 (30.8) | |
| | District Hospital | 35 (49.3) | 33 (14.5) | |
| | Total | 71 (100) | 227 (100) | |
| **Area of residence** | Rural | 16 (22.5) | 101 (44.5) | 0.00 |
| | Urban | 55 (77.5) | 126 (55.5) | |
| | Total | 71 (100) | 227 (100) | |
| **Type de health facilitie** | Private | 18 (25.4) | 97 (42.7) | 0.009 |
| | Public | 53 (74.6) | 130 (57.3) | |
| | Total | 71 (100) | 227 (100) | |
| **Sex of child** | Female | 38 (53.5) | 110 (48.5) | 0.46 |
| | Male | 33 (46.5) | 117 (51.5) | |
| | Total | 71 (100) | 227 (100) | |
| **Birth rank** | | | | 0.08 |
| | 1 | 29 (40.8) | 61 (26.9) | |
| | 2 | 13 (18.3) | 52 (22.9) | |
| | ≥3 | 29 (40.8) | 114 (50.2) | |
| | Total | 71 (100) | 227 (100) | |
| **Reason for the health facility visit** | Vaccination | 64 (90.1) | 210 (92.5) | 0.52 |
| | No vaccination | 7 (9.9) | 17 (7.5) | |
| | Total | 71 (100) | 227 (100) | |
| **Caregiver's age (years)** | <21 | 11 (15.5) | 20 (8.8) | 0.06 |
| | [21–31] | 10 (14.1) | 58 (25.6) | |
| | >31 | 50 (70.4) | 149 (65.6) | |
| | Total | 71 (100) | 227 (100) | |
| **Caregiver's education** | 0 | 0 (0) | 1 (0.4) | 0.39 |
| | 1 | 3 (4.2) | 24 (10.6) | |
| | 2 | 54 (76.1) | 161 (70.9) | |
| | 3 | 14 (19.7) | 41 (18.1) | |
| | Total | 71 (100) | 227 (100) | |

**Chi-square or Fisher test p-probability. Analysis was done on 298 children surveyed with vaccination cards that contain at least one vaccination date.

**c. Immunization practices by health personnel.** When a new booklet was issued at home, 87.04% (47/54) of health workers advised caretakers to keep it safe and 74.07% (40/54) asked to bring the booklet on all visits to the health facility (Table 5). However, only 55.56% (30/54) of the staff always told the janitors about the vaccines administered while giving advice in case any side effects occurred, regardless of the vaccine. When the janitor forgot the vaccination card at home, 46.30% (25/54) of the staff made the decision to give the caretaker a temporary card, vaccinate, and record the day's vaccinations in the vaccination log, while asking the janitor to bring the card on the next visit. When the caregiver stated that the immunization card was lost or damaged, 62.3% (40/54) of the staff handed out a new immunization card and transcribed all the vaccines received from the register (Table 5).

**Table 4. Knowledge of health personnel in the Dschang health district about vaccination in 2021.**

| Modalities | Numbers (N = 88) | % |
|---|---|---|
| **Health personnel who have been trained in VPD control during their basic training** | **76** | **86.36** |
| **Health personnel who received training (seminar) on the control of VPDs after their basic training** | **69** | **78.41** |
| **Knowledge of immunization schedule**: Number of health personnel who identified the listed VPDs (n = 88) | | |
| BCG | 88 | 100 |
| Measles-Rubella | 86 | 97.7 |
| Pentavalent (diphtheria, tetanus, pertussis, Hib, HepB) | 80 | 90.9 |
| Polio vaccine | 85 | 96.6 |
| Rotavirus | 76 | 86.4 |
| Pneumococcal vaccine (PCV) | 77 | 87.5 |
| Yellow fever | 69 | 82.1 |
| **Vitamin A** | **61** | **72.6** |
| **No vaccine** | **16** | **45.7** |
| **Contraindications to vaccination (all antigens) identified by health workers (n = 88)** | | |
| Local reaction to a previous dose | 27 | 30.7 |
| Mild fever | 42 | 47.7 |
| Epileptic seizures while on medication | 21 | 23.8 |
| **Pneumonia or other serious illness** | **36** | **40.9** |
| HIV+ child | 8 | 9.1 |
| None of the above | 17 | 19.3 |
| **Health workers reporting insufficient or outdated knowledge about vaccines and immunization (n = 88)** | 62 | 70.4 |
| **Health care workers reporting that filling out immunization records on immunization day would delay routine delivery of childhood immunization (n = 88)** | 25 | 28.4 |

**VPD**: Vaccine-Preventable Diseases; **BCG**: Bacillus Calmette and Guerin; **HepB**: Hepatitis B virus; **Hib**: Heamophylus Influenzae type B

## Discussion

The objective of this study was to determine the prevalence and characteristics of missed opportunities for vaccination among children aged 0–23 months in the Dschang Health District in 2021.

We used children's immunization records and/or records of immunization services to drive evidence. The study showed a rate of immunization cards possession of 97.32%, which is far above that found in MOV assessments in Chad (86%) and Malawi (63%) in 2015 [11] and Burkina Faso (90.26%) [12] as well as, in surveys conducted in Cameroon in 2011, and 2018 [10, 14]. what is quite interesting is the proportion not completely vaccinated (zero-dose), found as 18% in children, far below one survey conducted in the West region in 2014 [15]. In 2018 in Cameroon, the level of vaccine coverage achievement was unsatisfactory, with less than 80% coverage, with less than 10% of children not completely vaccinated [10].

The minimum interval between two vaccine doses should be respected to ensure the validity and effectiveness of the vaccine dose received by children [1]. In our study, DTP/HepB/Hib (Penta),which is the EPI tracer antigen, 15.54% of Penta 1 doses to 9.24% of Penta 3 doses, received early [1]. Although these results are different from those found in Burkina Faso in 2016 (early doses: 5% Penta 1 to 2% Penta 3 and 25% to 45% late doses), the evolution remains

**Table 5. Vaccination practice by health workers in the Dschang health district in 2021.**

| Modalities | Numbers (N = 54) | % |
|---|---|---|
| **Instructions given to caregivers when they receive a new immunization card for their child** | | |
| Keep the card safe | 47 | 87.0 |
| Bring the card with you on all visits to the health facility | **40** | **74.1** |
| Bring this card only when you come for immunizations | 16 | 29.6 |
| **Circumstances in which the HCP tells attendants which vaccines he or she is administering, while giving advice in case of APID** | | |
| Always, regardless of the vaccine used and the type of IPD that may occur | **30** | **55.6** |
| Only if the vaccine administered could produce an allergic reaction | 12 | 22.2 |
| Other * | 8 | 14.8 |
| **HCP decisions when it is determined that the child's caregiver r has forgotten the immunization card at home** | | |
| I give him a temporary card, vaccinate, record in the register and ask him to bring the old card at the next visit | **25** | **46.3** |
| I do not vaccinate the child and ask the mother to come back with the card next time | 9 | 14.8 |
| I give the child a new card. vaccinate and record that day's vaccinations on the new card and in the register | 8 | 14.8 |
| Other ** | 12 | 22.2 |
| **HCP decisions when a caregivers says card was lost or damaged** | | |
| I give a new card and transcribe all previous immunizations from the logbook | **40** | **62.3** |
| I give him a new card and record all future immunizations on the new card | 8 | 14.8 |
| Other *** | 6 | 11.1 |

**HCP**: Health care personnel; **APID**: Adverse Events Post Immunization; **Other*** (Never, since this information may be counterproductive and discourage participation in the immunization program + The likelihood of an adverse event related to immunization occurring is so low that I rarely need to provide this information + Only when the mother or caregiver requests this information); **Other**** (I give her a new card, vaccinate and record previous vaccinations + I vaccinate without the replacement card, but record it in the registry and on the card the next time I vaccinate + I vaccinate without the replacement card, but record it in the registry only); **Other***** (I vaccinate without the replacement card, record it in the registry only + I give her a new card and ask the mother to tell me all previous vaccinations so I can record them).

similar in both cases with the number of early doses decreasing from the first to the third Penta dose and the number of late doses increasing instead [12]. Overall vaccine promptness was 73.5% for all EPI vaccines and ranged from 73.7% in Penta 1 to 66.7% in Penta 3. These results followed the same trend as those found in Burkina Faso in 2016 (from 67% Penta 1 to 35% Penta 3) although they were notably higher [12]. This finding explains the very high prevalence of MOVs (60.33%) obtained in children aged 0–23 months in the district. Notwithstanding the fact that it is high, it is still lower than that found in Chad (77%) and Malawi (92%) in 2015 [11] and Burkina Faso in 2016 (76%) [12]. In contrast to the results found in the Dominican Republic (43.7%) [16]. Bangui (33%) [6], Brazzaville (12.5%) [17] and CAR (25% to 31%) [18]. this prevalence remains very high. The prevalence of MOVs in the district was quite below the national estimate of 75% found in 2018 [10], and could not offer the appropriate picture to tackle associated factors.

The results of the KAP surveys of health personnel provide ample information on the likely causes of EPI. Nearly 70.45% of health workers reported that their knowledge about vaccination was insufficient or outdated and 72.62% had identified vitamin A as an EPI vaccine (data not tabulated). These results corroborate with those obtained in Burkina Faso in 2016 [12].

MOVs may in some cases occur because daily attitudes are not conducive for them to be detected and reduced as more than 78.41% of the staff felt that the immunization status of children should only be assessed by the staff responsible for immunization. In particular, the rate of possession of the booklets remains very high and could be justified by the fact that in daily practice, nearly ¾ of the staff, when handing out a new vaccination card, ask the accompanying staff to bring it to any visit to the health facility, regardless of the reason for the visit. Refresher training and updated EPI supervision in the Dschang health district will therefore be necessary to improve the knowledge and attitudes of health personnel, which will have a positive impact on immunization coverage.

In addition to the poor integration of immunization services and weak logistics, including vaccine shortages, the absence of daily immunization services in almost all health facilities has also contributed to the occurrence of MOVs. Unfortunately, current practices deeply rooted in the routine of many health facilities aim at minimizing vaccine wastage and health workers are reluctant to vaccinate some children visiting the health facilities when they are late [1]. The situation of early and late doses is related to the reluctance or refusal of health workers to open a multidose vial unless a critical mass of children is gathered especially for lyophilized vaccines such as BCG, RR and VAA [1, 19]. Packaging of single-dose vials or prefilled syringes would be a possible approach to address this problem. although it is difficult to remedy [20]. Similarly, a paradigm shift would be needed from a focus on "wastage rate" to "utilization rate" of vaccines in the international community.

Although the present study demonstrated that MOV is present in the district, the study had some limitations: the calculation of MOV prevalence accounts only for records with vaccination dates, hence ignoring those from recall and tick mark. Indeed, these later vaccination sources are somewhat subject to information bias. In addition, because of very low proportion of MOV, multivariate logistic regression would have not be informative. However, investigation is needed to explore factors that influence the burden of MOV.

## Conclusion

Vaccination timeliness in the Dschang health district was low, and the prevalence of MOV was non-neglected. Further study is needed to identify potential strategies to reduce missed opportunities for vaccination.

## Supporting information

**S1 Fig. Allocation of selected health facilities by type in the DHS during the cross-sectional survey in 2021.**
(TIF)

**S1 Table. Source of immunization among children aged 0–23 months in the DHD by doses of vaccine received in 2021.**
(DOCX)

**S2 Table. Vaccination timeliness among children with cards seen and dates in the DHD in 2021.**
(DOCX)

**S3 Table. Socio demographic information of health workers from surveyed health facilities in the DHS in 2021.**
(DOCX)

**S4 Table. Attitudes of health personnel in the Dschang health district towards immunization in 2021.**
(DOCX)

**S1 Data. Sample data used to compute vaccination timeliness and missed opportunities indicators in the DHS in 2021.**
(XLSX)

**S1 File. R code used to compute vaccination timeliness indicators.**
(R)

## Acknowledgments

The authors wish to thank NDOMBOL Ivan ZACHEE Gottlieb (Hope For Nations (HFN)), BEBE MOUSSANGO ENANGUE Blanche Lydie (Hope For Nations (HFN)) and NWUA-BUEZE Félix for their great support to this work. The authors wish to also thank all District health service, health facility medical staff and caregivers for their participation in this assessment. Finally, our appreciation goes to the many caregivers and health workers who participated in the interviews and freely provided us with their valuable time and their opinions.

## Author Contributions

**Conceptualization:** Arsène Gautler Dombou Zeufack, Solange Whegang Youdom.

**Data curation:** Arsène Gautler Dombou Zeufack.

**Formal analysis:** Arsène Gautler Dombou Zeufack, Solange Whegang Youdom.

**Investigation:** Arsène Gautler Dombou Zeufack.

**Methodology:** Arsène Gautler Dombou Zeufack, Solange Whegang Youdom.

**Software:** Arsène Gautler Dombou Zeufack.

**Supervision:** Jérôme Ateudjieu.

**Validation:** Solange Whegang Youdom, Jérôme Ateudjieu.

**Writing – original draft:** Arsène Gautler Dombou Zeufack, Solange Whegang Youdom.

**Writing – review & editing:** Arsène Gautler Dombou Zeufack, Diomede Noukeu Njinkui, Solange Whegang Youdom.

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
