## [Decision Letter · Decision Letter 0]

28 Dec 2022

PGPH-D-22-01589

Missed opportunities for vaccination and associated factors among children aged 0 to 23 months in Dschang health district, West region, Cameroon: a cross-sectional survey

Dear Dr. Dombou Zeufack,

Thank you for submitting your manuscript to PLOS Global Public Health. After careful consideration, we feel that it has merit but does not fully meet PLOS Global Public Health’s publication criteria as it currently stands. Therefore, we invite you to submit a revised version of the manuscript that addresses the points raised during the review process.

The reviewer comments were relatively minor. Please let us know if you do not agree with any comments, but if you do, please indicate in your response where exactly you have made changes in the manuscript.

We look forward to receiving your revised manuscript.

Kind regards,

Abram L. Wagner, PhD, MPH

Academic Editor

Journal Requirements:

a. Please clarify all sources of funding (financial or material support) for your study. List the grants (with grant number) or organizations (with url) that supported your study, including funding received from your institution. 

b. State the initials, alongside each funding source, of each author to receive each grant.

c. State what role the funders took in the study. If the funders had no role in your study, please state: “The funders had no role in study design, data collection and analysis, decision to publish, or preparation of the manuscript.”

d. If any authors received a salary from any of your funders, please state which authors and which funders.

2. We do not publish any copyright or trademark symbols that usually accompany proprietary names, eg (R), (C), or TM  (e.g. next to drug or reagent names). Please remove all instances of trademark/copyright symbols throughout the text, including R on page 4 and TM on page 3.

4. In the online submission form, you indicated that [Insert text from online submission form here]. All PLOS journals now require all data underlying the findings described in their manuscript to be freely available to other researchers, either 1. In a public repository, 2. Within the manuscript itself, or 3. Uploaded as supplementary information.

Additional Editor Comments (if provided):

Reviewers' comments:

Reviewer's Responses to Questions

**Comments to the Author**

1. Does this manuscript meet PLOS Global Public Health’s publication criteria? Is the manuscript technically sound, and do the data support the conclusions? The manuscript must describe methodologically and ethically rigorous research with conclusions that are appropriately drawn based on the data presented.

Reviewer #1: Partly

Reviewer #2: Yes

2. Has the statistical analysis been performed appropriately and rigorously?

Reviewer #1: Yes

Reviewer #2: Yes

3. Have the authors made all data underlying the findings in their manuscript fully available (please refer to the Data Availability Statement at the start of the manuscript PDF file)?

Reviewer #1: No

Reviewer #2: Yes

4. Is the manuscript presented in an intelligible fashion and written in standard English?

Reviewer #1: Yes

Reviewer #2: Yes

5. Review Comments to the Author

Reviewer #1: Introduction

There are various typos in the introduction

"Indeed, it prevents nearly 75,000 children from suffering serious physical, mental or neurological disabilities [1]" this number doesn't seem correct . It is too low.

What is RR1 ?

The second last paragraph of introduction is more suited for discussion

Methods

What is DHD?

what is meant by couples/children ? is there a better terminology that can be used ?

What is WHO quota sampling ?

What is CAP survey ?

Results

One decimal point should be enough

Please don't start sentences with numbers

Results are too long . some of the tables can be moved to appendix.

Conclusion needs to be crisp and not more than 2-3 sentences

Reviewer #2: The manuscript numbering system was not done making review process difficult

Page 1

The title is topical as the EPI vaccination schedule in in developing countries and completion of RI is still a problem and the needs for more studies is appropriate

Abstract

....in the west region

MOV prevalence range from 69.7%(19/29) to 98.5%(330/335)

...74%asked parents

... strategies for remedying this includes

Page 2

....remove expanded program on immunization after the ref

In Cameroon vaccination has been offered throughout the country

Table2: Types of vaccination...

Table 5: Sociodemographic information of health workers...

Table 9: Logistic regression of MOV among children...

Discussion

Page 16

....

6. PLOS authors have the option to publish the peer review history of their article (what does this mean?). If published, this will include your full peer review and any attached files.

**Do you want your identity to be public for this peer review?** For information about this choice, including consent withdrawal, please see our Privacy Policy.

Reviewer #1: No

Reviewer #2: No

---

## [Editor Report · Decision Letter 1]

3 Apr 2023

PGPH-D-22-01589R1

Timeliness and Missed opportunities for vaccination among children aged 0 to 23 months in Dschang health district, West region, Cameroon: a cross-sectional survey

Dear Dr. Whegang Youdom,

Thank you for submitting your manuscript to PLOS Global Public Health. After careful consideration, we feel that it has merit but does not fully meet PLOS Global Public Health’s publication criteria as it currently stands. Therefore, we invite you to submit a revised version of the manuscript that addresses the points raised during the review process.

We look forward to receiving your revised manuscript.

Kind regards,

Abram L. Wagner, PhD, MPH

Academic Editor

Journal Requirements:

Additional Editor Comments (if provided):

Can you go through the results and change it to only one decimal place being used?
---

## [Editor Report · Decision Letter 2]

8 May 2023

Timeliness and Missed opportunities for vaccination among children aged 0 to 23 months in Dschang health district, West region, Cameroon: a cross-sectional survey

PGPH-D-22-01589R2

Dear Mrs Whegang Youdom,

We are pleased to inform you that your manuscript 'Timeliness and Missed opportunities for vaccination among children aged 0 to 23 months in Dschang health district, West region, Cameroon: a cross-sectional survey' has been provisionally accepted for publication in PLOS Global Public Health.

Best regards,

Abram L. Wagner, PhD, MPH

Academic Editor